# Curve Fitting for Damage Evolution through Regression Analysis for the Kachanov–Rabotnov Model to the Norton–Bailey Creep Law of SS-316 Material

**DOI:** 10.3390/ma14195518

**Published:** 2021-09-23

**Authors:** Mohsin Sattar, Abdul Rahim Othman, Maaz Akhtar, Shahrul Kamaruddin, Rashid Khan, Faisal Masood, Mohammad Azad Alam, Mohammad Azeem, Sumiya Mohsin

**Affiliations:** 1Department of Mechanical Engineering, Universiti Teknologi PETRONAS, Seri Iskandar 32610, Perak, Malaysia; shahrul.k@utp.edu.my (S.K.); azadalam.mech3@gmail.com (M.A.A.); mazeem.me@zhcet.ac.in (M.A.); 2Department of Mechanical Engineering, NED University of Engineering & Technology, Karachi 75270, Sindh, Pakistan; maaz@neduet.edu.pk (M.A.); sumiya.mohsin@yahoo.com (S.M.); 3Mechanical Engineering Department, College of Engineering, Imam Mohammad Ibn Saud Islamic University, Riyadh 11432, Saudi Arabia; rakhan@imamu.edu.sa; 4Department of Electrical Engineering, Universiti Teknologi PETRONAS, Seri Iskandar 32610, Perak, Malaysia; fslmsd@gmail.com

**Keywords:** creep deformation, regression analysis, Kachanov–Rabotnov model, damage evolution, creep rupture

## Abstract

In a number of circumstances, the Kachanov–Rabotnov isotropic creep damage constitutive model has been utilized to assess the creep deformation of high-temperature components. Secondary creep behavior is usually studied using analytical methods, whereas tertiary creep damage constants are determined by the combination of experiments and numerical optimization. To obtain the tertiary creep damage constants, these methods necessitate extensive computational effort and time to determine the tertiary creep damage constants. In this study, a curve-fitting technique was proposed for applying the Kachanov–Rabotnov model into the built-in Norton–Bailey model in Abaqus. It extrapolates the creep behaviour by fitting the Kachanov–Rabotnov model to the limited creep data obtained from the Omega-Norton–Bailey regression model and then simulates beyond the available data points. Through the Omega creep model, several creep strain rates for SS-316 were calculated using API-579/ASME FFS-1 standards. These are dependent on the type of the material, the flow stress, and the temperature. In the present work, FEA creep assessment was carried out on the SS-316 dog bone specimen, which was used as a material coupon to forecast time-dependent permanent plastic deformation as well as creep behavior at elevated temperatures and under uniform stress. The model was validated with the help of published experimental creep test data, and data optimization for sensitivity study was conducted by applying response surface methodology (RSM) and ANOVA techniques. The results showed that the specimen underwent secondary creep deformation for most of the analysis period. Hence, the method is useful in predicting the complete creep behavior of the material and in generating a creep curve.

## 1. Introduction

In materials, creep deformation can be divided into three stages: primary, secondary, and tertiary [1]. At low temperatures and modest stresses, the primary creep regime dominates. Primary creep strain accumulates in high-temperature alloys, but it is generally undetectable in studies. At a low-to-intermediate stress and temperature, the secondary creep regime dominates [2]. This is the most stable domain, in which a balance of strain-hardening and recovery mechanics makes creep deformation prediction simple. The tertiary creep regime dominates at intermediate to high temperatures and stresses. The non-linear accumulation of creep-damage ruptures, which contributes to massive creep deformation, characterizes this regime.

Since the advent of continuum damage mechanics (CDM) in 1967–68, major effort has gone into applying CDM to the assessment of creep damage in high-temperature components, as described by Chaboche [3]. According to Murakami [4], the CDM approach has been utilized to solve high-temperature engineering challenges in the aerospace, nuclear, and power-generating industries. Engineers can utilize CDM-based creep constitutive models to anticipate not only the material’s constitutive reaction but also the material’s subsequent rupture through damage evolution. As Betten [5] researched, the CDM system has previously been used to model elastic–brittle, elastic–plastic, fatigue, creep, creep–fatigue, iso-thermal and iso-thermomechanical, anisotropic, corrosion, and irradiation-induced damage. The CDM effective stress concept can be used to characterize the damage process in materials from crack initiation through ruptures. A continuous damage variable is connected with the constitutive model’s viscosity function in the CDM technique to incorporate the impacts of microstructural damage into the constitutive response. As documented by Kachanov [6], the damage variable is expected to grow from zero (no damage) to unity (rupture).

According to Skrzypek [7], within CDM, the smallest volume statistically representative of the mean constitutive response including a representative number of micro-heterogeneities is called a representative volume element (RVE). The effective stress concept allows transformation from the physical space of the heterogeneously damaged RVE to an effective space of homogeneous undamaged RVE including damage through effective stress increase. Damage is an irreversible representation of heterogeneous micro-processes that occur during the deformation of a material, and its distribution and evolution are influenced by strain history, boundary conditions, time, and the environment. While the physical damage is difficult to quantify, Gordon and Stewart [8] examined it and found that it may be defined as a reduction in area due to internal and surface defects. The effective stress notion may be analytically inferred using this concept to obtain the net/effective stress, as shown in Equation (1):(1)$$\bar{\sigma }=\sigma \times \frac{A_{0}}{A_{net}}=\frac{\sigma }{\left(1-\frac{A_{0}-A_{net}}{A_{0}}\right)}=\frac{\sigma }{\left(1-\omega \right)}$$
where $A_{net}$ is the current area, $A_{0}$ is the initial area, $\bar{\sigma }$ is the equivalent stress, *σ* is the net/effective stress, and *ω* represents the damage or reduction in area. Physical damage is replaced with an increase in the applied stress, which is more effective [9]. By equating finite element methods to RVE and using the effective stress concept, the CDM theory can be implemented into FEM codes.

Several material models have been developed since 1929 to provide predictions for creep behavior of the materials. These models were developed under certain boundary conditions, and, for that, the models may possess some limitations, as explained by Yao et al. [10]. Three established models are discussed in this article, and a method is being devised to predict the material’s creep response and behavior in the tertiary stage through a damage evolution parameter [11]. Starting with the Norton–Bailey (NB) model, also known as Norton’s power law, it is integrated in finite element Abaqus software and is widely used for creep analysis in combination with other models developed by Bailey and Norton [12]. The NB model works as a benchmark for the development of other models. The creep deformation behavior of materials displaying time-dependent, inelastic deformation can be predicted using the model. It is useful in predicting material’s creep behavior in the secondary creep regime. On the other hand, the Omega model developed by the Material Properties Council offers a good prediction model; it is widely used due to its simplicity and its lower dependency on material constants. Prager [13] proposed the model in 1995, and it is a well-documented creep evaluation process with an excellent track record for associated property relations covering a wide range of materials. The Omega model is a method for calculating the remaining life of a component that is working in the creep zone at high temperatures and pressures. According to Yeom et al. [14], a strain rate parameter and a multi-axial damage parameter are used to forecast the rate of strain accumulation, creep damage accumulation, and the remaining time to failure as a function of the stress state and the temperature. It models primary and secondary creep regime deformations and is good in predicting material’s rupture time at lower temperatures [15]. The other important model is the Kachanov–Rabotnov model, which is one of the earliest implementations of the continuum damage mechanics (CDM) approach for creep, as proposed by Kachanov and Rabotnov [6]. Secondary and tertiary creep deformation can be modelled using the set of coupled equations. Significant efforts have been made to improve the KR law, and variations of the KR law have been developed in recent years to generate contour deformation maps [8]. To define transversely isotropic creep damage properties and to estimate stress-independent tertiary creep damage constants using both strain- and damage-based analytical approaches, methods are being devised by Stewart and Gordon [16]. Another significant creep prediction model is theta projection, which can be employed under creep conditions. Evans and Wilshire [17] presented the theta projection model in 1985 to predict multistage creep deformation (primary, secondary, and tertiary). A modified form of the theta projection model, which is specific for steel sheets, was investigated and applied by Alipour and Nejad [18] to ferritic steel alloys. The hyperbolic model, developed by Stewart in 2013 [19], is the latest development to creep prediction models, which was effectively utilized by Alipour and Nejad [20] for ferritic steels at elevated temperatures. The three-stage creep damage model has the ability to simulate primary, secondary, and tertiary creep with better accuracy.

Several researchers found that each creep prediction model has proved to be accurate for specific materials under certain stress levels and temperature conditions, but neither single model can precisely predict the creep behavior for variations of material alloys nor can any model meet expectations for service conditions as investigated by Benallal [21]. For example, the Norton–Bailey model only models the secondary creep regime with no prediction for the other regimes. It produced an overall error when primary and tertiary creeps were dominant. Batsoulas [22] highlighted the limitations of the Norton–Bailey creep model by arguing that the use of this relation in the design indicated that (i) the creep curve is a straight line, (ii) primary and tertiary creeps are neglected, and (iii) the rate of secondary creep is defined as the exclusive designing parameter, which is definitely not the case. According to May and Furtado [23], material constants in the Norton–Bailey equation are heavily dependent upon temperature, and, because of that, the model is unable to produce accurate results at higher temperatures.

The fracture strain in the MPC Omega model is difficult to estimate for the equipment under service conditions due to limited temperature-dependent materials data at elevated temperatures. As pointed out by Maruyama et al. [24], the model also lacks any methods of indicating prior and ongoing damage in the material where the fracture begins and will most likely expand. The model was also unable to predict accurate creep results for high-temperature and high-stress applications, which highlights its major drawback. In the case of creep life prediction for dissimilar *P91/12Cr1MoV*-steel-welded joints, as investigated by Chen et al. [25], the accuracy of the predicted rupture time became much closer to the actual values at the lower stress level. The error was 12.7% at a stress of 160 MPa, and it increased to 75.4% at a stress of 220 MPa under the same temperature conditions of 550 °C.

The model proposed by Kachanov–Rabotnov (KR) is promising, but it involves a large number of material constants, and the formulation does not consider the primary creep regime for the analysis. KR emulates both continuum creep damage and discontinuous plastic damage at rupture within a continuous function, in which the model complexities have proven difficult for integration in FE analysis [26]. The difficulty in finding the tertiary creep damage constants is another key drawback of the KR model. Haque and Stewart [27] conducted a comparative analytical case study of sine-hyperbolic (sin-h) and Kachanov–Rabotnov (KR) creep damage models in forecasting the minimum creep strain rate, creep deformation, damage progression, and the rupture of 304 stainless steel at 600 °C and 700 °C. Because the KR rupture predictions are linear on a log–log scale, they cannot adequately explain the sigmoidal behavior seen in the experimental data. The KR-model can be re-calibrated to meet either high- or low-stress environments, but it cannot model both at the same time. At the yield strength, the sin-h rupture predictions approach a value less than unity but close to the nominal ultimate tensile strength of 304SS. As the rupture time approached zero, the KR rupture prediction approached an accuracy value of 1.35 times greater than the ultimate tensile strength [27]. Haque and Stewart [28] and Stewart and Gordon [29] eventually identified the faulty character of the KR model and named it as a “brittle curve” phenomenon. The KR model was adjusted to handle the high stress to low stress bend by adding extra factors and material constants while keeping the flaw that critical damage is less than unity. In brief, those models have dealt with the issue of nucleation and plasticity for crack initiation and growth at some success level, but limitations in accurately detecting and identifying creep crack growth are still apparent [30].

Little attention has been paid to designing optimization and sensitivity analyses involving all design parameters for creep prediction, especially in the tertiary creep stage [31]. In this article, a curve-fitting technique is proposed for applying the Kachanov–Rabotnov model into the embedded Norton–Bailey FEA model. The creep parameters involving the damage evolution parameter are regressed to represent a material’s behavior in the tertiary creep stage. The extrapolation of creep behavior has been performed by fitting different forms of creep models, Kachanov–Rabotnov in this case, with a small number of creep data sets and then simulating beyond the available data points, as explained by Brown et al. [32]. Comparative assessment of the proposed Kachanov–Rabotnov model curve fitting-technique was then preformed with other creep damage models, Omega and Norton–Bailey, to highlight its significances.

The purpose of the study was to identify tertiary creep damage constants, which are difficult to determine for the materials at high temperatures. This research has significance as it provides a method to determine the tertiary stage constants and to analyze material’s behavior at the tertiary creep stage. The extrapolations were based on the assumption that the existing trend in the available data set will continue and that the model’s functional form appropriately captures the creep behavior in the extrapolated region. Extrapolations are generally unreliable and difficult to assess [33]. The 0% and 100% dependability bands for creep–rupture expand when the applied stress is lowered, to the point that the bands may differ decades apart at low stress levels and elevated temperatures [34]. The uncertainty of extrapolations paired with material performance necessitates the use of generous safety factors in design. There is no guarantee of reliability for a new material with 30,000 h of creep data extrapolated to 100,000 h [28].

In this case study, FEA creep assessment was performed on the material SS-316 dog bone specimen, which was considered as a material coupon to predict time-dependent plastic deformation along with the creep behavior at an elevated temperature of 650 °C and under constant stresses [35]. A data-optimization and sensitivity study was conducted by applying response surface methodology along with the ANOVA technique to track the material’s creep behavior for the three creep damage models [36]. The results indicated that the specimen underwent steady secondary creep deformation for most of the analysis period. Hence, the method is useful in predicting the complete creep behavior of the material and in generating a complete creep curve covering the primary, secondary, and tertiary creep stages. The uncertainty of the material’s behavior in the tertiary stage near to rupture can be predicted accurately [37].

## 2. Theoretical Framework

This section describes the theoretical framework of the creep material models, their mathematical formulation, and their combined implementation for the prediction of a material’s creep behavior, SS-316 in this case.

### 2.1. Norton Bailey Model

The most famous and common minimum creep strain rate law is Norton’s power law (1929), which was based on the Arrhenius rate equation [38] and which is depicted in Equation (2). (2)ε˙cr=B′σnexp−QcRTtm
where ${\dot{\varepsilon }}_{cr}$ represents the minimal creep strain rate, *B*′ represents the material constant, *σ* is the applied stress, *n* represents the power law exponent, $Q_{c}$ represents the activation energy, *R* is the universal gas constant, and *T* represents the applied temperature. Equation (2) is further simplified at a constant temperature, taking the form of Equation (3).
(3)$${\dot{\varepsilon }}_{cr}=A\;\sigma^{n}t^{m}$$
without considering time, Equation (3) becomes:(4)$${\dot{\varepsilon }}_{cr}=A\sigma^{n}$$
where,
(5)A=B′exp−QcRT

It is worth noting that *A*, *n*, and *m* are temperature-dependent material constants that are not affected by stress. While *n* and *m* are unitless, *A* has units that are consistent to those of time *t* and stress *σ*. The time-differentiated version of Equation (3) is often referred to as the time-hardening formulation of the power-law creep [39].

### 2.2. Omega Model

In the Omega model, the life fraction used (or damage parameter, *ω*) is defined as in Equation (6):(6)$$\omega =\frac{t}{t_{r}}=\frac{\dot{\varepsilon }\Omega t}{1+\dot{\varepsilon }\;\Omega t}$$
where *t* is the current time, *t_r_* is the rupture life, $\dot{\varepsilon }$ is the current creep strain rate, and *Ω* is the material creep damage constant. When the current time approaches the rupture, then (*t* ▶ *t_r_* & *t*/*t_r_* ▶ 1) and Equation (6) collapses. Thus, the life fraction evolves from zero to near unity (0 ≤ *t*/*t_r_* < 1). Prager [15] has proposed that the creep strain will have the following relation as in Equation (7):(7)1−ε˙0Ωt=1eε˙Ω
where $\dot{\varepsilon }$ is the creep strain, ${\dot{\varepsilon }}_{0}$ is the initial creep strain rate constant, and *Ωt* is the damage constant with respect to time. Taking the natural logarithm on both sides and rearranging gives Equation (8):(8)$$\dot{\varepsilon_{C0}}=\frac{-1}{\Omega }\;\ln\left(1-{\dot{\varepsilon }}_{0}\Omega t\right),$$

Equation (8) can be further simplified by taking derivatives with respect to time and replacing (1 − ${\dot{\varepsilon }}_{0}$ *Ωt*) as follows:(9)$$\dot{\varepsilon_{c}}={\dot{\varepsilon }}_{c0}\;e^{\varepsilon_{c}\Omega }$$

The Omega model, as can be shown from Equation (9), assumes that the creep strain rate is proportional to the exponent of the accumulated creep strain and ignores the primary creep stage. The Omega model is easy to apply as it only requires two constants: *ε*_0_ and *Ω* [24]. The constant *Ω* can be obtained from experimental data using Equation (7). The large value of *Ω* indicates that most of the life spent by the material is at a low creep strain rate followed by a rapid increase in the creep strain rate before failure. A low value of *Ω* indicates that most of the time spent is in the tertiary creep regime [40]. The initial strain rate *ε*_0_ and the omega (*Ω*) values are dependent on stress and temperature and can be expressed as a creep power law in Equations (10) and (11):(10)$$\varepsilon_{0}=A_{0}\sigma^{n_{0}}exp\left(-\frac{Q_{0}}{RT}\right),$$
(11)$$\Omega =A_{\Omega }\sigma^{n_{\Omega }}exp\;\left(-\frac{Q_{\Omega }}{RT}\right).$$
where *A*_0_ and *A_Ω_* are stress coefficients, *n*_0_ and *n_Ω_* are stress exponents, *Q*_0_ is the apparent activation energy, and *Q_Ω_* is the value indicating the temperature dependence of *Ω* [24]. It can be seen that the magnitude of the *Ω* value increases with the decrease in both stress and temperature, which corresponds to the descriptions of Ohgeon [41].

### 2.3. Kachanov–Rabotnov Model

The classic Kachanov–Rabotnov law consists of the coupled strain rate and damage evolution equations, which are as follows in Equations (12) and (13) [29].
(12)$$\dot{\varepsilon }=A(\frac{\sigma }{1-\omega }{)}^{n}$$
(13)$$\dot{\omega }=\frac{M\;\sigma^{x}}{{\left(1-\omega \right)}^{\emptyset }},0<\omega <1$$
where *M*, *χ*, and ∅ are tertiary creep damage constants, and the creep strain is similar to Norton’s power law for secondary creep with the same *A* and *n* as secondary creep constants. Isochoric creep behavior is assumed, and the secondary and tertiary creep damage constants can be calculated analytically [42]. Equations (14) and (15) were obtained by taking Equation (13) and conducting a separation of variables, indefinite integration, and simplifications.
(14)$$t(\omega ,\sigma )=\frac{1-{\left(1-\omega \right)}^{\emptyset +1}}{\left(\emptyset +1\right)M\;\sigma^{x}},$$
(15)$$\omega (t,\sigma )=1-{\left[1-\left(\emptyset +1\right)M\;\sigma^{x}\;t\right]}^{\frac{1}{\emptyset +1}},$$
where *t* is the current time and *ω* represents the current damage. These equations can be used to calculate the current time from the current damage and stress, or the current damage from the current time and stress. Stewart and Gordon [16] developed two useful techniques for analyzing and implementing the Kachanov–Rabotnov model: the strain approach (SA) and the damage approach (DA). The techniques are implemented depending on the material’s analysis for secondary and tertiary creep stages.

## 3. Methodology

This section explains the methodology adapted for implementing the curve-fitting method for creep prediction by the combination of creep models through regression and their comparison.

### 3.1. Analytical Creep Strain

For finding creep strain and the creep strain rate, the research study is divided into two sections. First, the creep strain was calculated analytically, using the creep material model from the Omega model formulation based on ASME FFS-1/API-579 standards [43] for the SS-316 dog bone specimen. Secondly, creep strains were calculated through FEA simulation in Abaqus using material model properties obtained from ASME BPVC section II part D, sub-part 2 standards [44]. Regression analysis was also implemented to obtain the creep parameter (*A*) and the stress exponent (*n*) for the Omega-Norton–Bailey and Kachanov–Rabotnov–Norton–Bailey regression models for the material coupon SS-316 in Abaqus. The results were later compared and validated with the experimental creep tests, as conducted by Christopher et al. [45].

The ASME FFS-1/API-579 standards [43] are designed to assist in the material’s data selection for the fitness-for-service assessment (FFS) of creep damage. It covers situations involving creep damage and flaws encountered in equipment exposed to service conditions for long periods of time, under high temperatures and pressures. The MPC Omega creep data are material-dependent, and a wide range of materials were described in ASME FFS-1/API-579 ranging from carbon steel to stainless steels and alloys. In this research study and analysis, material type SS-316 was selected. The coefficients in Table 1 are the estimates of typical material behavior used in the analytical calculations. Coefficient values were derived after extensive examining of the material behavior and were obtained from ASME FFS-1/API-579 standards [43].

The parameters and formulations are defined in ASME FFS-1/API-579 standards specifically for the MPC Omega model. It should be emphasized that the MPC Omega model does not take into account the impacts of primary creep. When the stress from the applied load is less than or equal to 50% of the minimum yield strength at the assessment temperature [12], the effect becomes negligible. The analytical creep strain for dog bone specimen, material coupon SS-316 was calculated using the following closed-loop Equations (16)–(22) taken from the ASME FFS-1/API-579 standards [43].
(16)$$\log_{10}\left[\dot{\varepsilon_{c0}}\right]=-[(A_{0}+{∆}_{\Omega }^{sr})+(\frac{A_{1}+A_{2}\;\;S_{1}+A_{3}S_{1}^{2}+A_{4}\;\;S_{1}^{3}}{T_{ref}+T})],$$
(17)$$\Omega_{m}=\Omega_{n}^{\delta_{\Omega +1}}+\alpha_{\Omega }+n_{BN},$$
(18)$$\log_{10}\left[\Omega \right]=[(B_{0}+{∆}_{\Omega }^{cd})+(\frac{B_{1}+B_{2}\;\;S_{1}+B_{3}S_{1}^{2}+B_{4}\;\;S_{1}^{3}}{T_{ref}+T}),$$
(19)$$\delta_{\Omega }=\beta_{\Omega }.(\frac{\sigma_{1}+\sigma_{2}+\sigma_{3}}{\sigma_{e}}-1),$$
(20)$$n_{BN}=-(\frac{A_{2}+2A_{3}\;\;S_{1}+3\;A_{4}S_{1}^{2}\;\;}{T_{ref}+T}),$$
$$T_{ref}=460\;\mathrm{for}\;{}^{\circ}F,\;\;\;\;T_{ref}=273\mathrm{for}\;{}^{\circ}C$$
(21)$$S_{1}=\log_{10}\left[\sigma_{e}\right],$$
(22)$$\sigma_{e}=\frac{1}{\sqrt{2}}\;{\left[{\left(\sigma_{1}-\sigma_{2}\right)}^{2}+{\left(\sigma_{1}-\sigma_{3}\right)}^{2}+{\left(\sigma_{2}-\sigma_{3}\right)}^{2}\right]}^{1/2}.$$

### 3.2. Finite Element Simulation—The Regression Model

The creep assessment model defined in the API 579-1/ASME FFS-1 standards use the MPC Omega model for inelastic analysis. Regression analysis proved to be an important statistical tool for estimating the relationships between dependent and independent variables, and the curve fitting may offer an effective means for reducing the number of datasets [46]. There are two approaches to lessen experimental creep deformation and data extracted from a normal creep experiment. The first method, known as the time-hardening formulation, involves maintaining constant time increments and monitoring strain at each point across different loads. The second technique, known as the strain hardening formulation, includes measuring the time it takes to reach the desired set of strain increments [39]. Figure 1 provides an overall process flow methodology to obtain the creep data based on the Omega-Norton–Bailey Regression Model [13] and the development of the dog bone specimen in Abaqus. The extracted data were then later input in Abaqus with other parameters for running the FEA simulation and for obtaining the creep and plastic strain results [47].

The details of the procedures are provided in the following steps [48]:

(i) The MPC Omega creep strain data were generated from Equation (9) discussed above for the Omega model. The strain rate from the MPC Omega model was stress- and temperature-dependent [24].

(ii) By considering the temperature as a constant, the strain rate can be calculated. The Norton–Bailey creep in Equation (4) discussed above was implemented for calculating the strain rate. Since the Norton–Bailey model is based on the creep power law, the equation for a general non-linear power law regression fit can be used.

(iii) By comparing the general power law regression equation with the Norton–Bailey power law, curve fitting can be executed for varying stresses and at different temperatures [39]. Extrapolation assumes that the current trend in the material’s behavior prevails over a period of time for the analysis [33]. The equation for general power law regression is represented in Equation (23) as:(23)$$y=A^{\prime }x^{B}$$
where *y* is the criterion variable and prediction response, *A′* is the curve coefficient, and *B* is the exponent of *x*, the predictor variable, which was compared with the Norton–Bailey power law regression in Equation (4). The parameter *B*, in Equation (24a), was compared with the derived stress exponent *n′*, in Equation (24b), for creep.
(24a)$$B=\frac{n^{\prime }\sum \left(\ln x\ln y\right)-\sum \left(\ln x\right)\sum \left(\ln y\right)}{n^{\prime }\sum [\left(\ln x{)}^{2}\right]-[(\sum \left(\ln x{)}^{2}\right]}$$
(24b)$$n=\frac{n^{\prime }\sum \left(\ln\sigma \ln\;\dot{\varepsilon }\right)-\sum \left(\ln\sigma \right)\sum \left(\ln\dot{\varepsilon }\right)}{n^{\prime }\sum [\left(\ln\sigma {)}^{2}\right]-[(\sum \left(\ln\sigma {)}^{2}\right]}$$

Similarly, the curve coefficient *A′*, in Equation (25a), was compared with the Norton–Bailey creep parameter *A*, in Equation (25b), for finding the parameters.
(25a)$$A^{\prime }=e^{\frac{\sum \left(\ln y\right)-B\sum \left(\ln x\right)}{n^{\prime }}}$$
(25b)$$A=e^{\frac{\sum \left(\ln\dot{\varepsilon }\right)-n\sum \left(\ln\sigma \right)}{n^{\prime }}}$$
where *x* and *σ* are independent variables, *y* and $\dot{\varepsilon }$ are dependent variables, and *n′* represents the number of samples. For the regression analysis [49], the stress is an independent variable, whereas the strain rate is a dependent variable. The stress range was set based on the accuracy of regression. A better curve fit can be obtained with a larger stress range of the sample data.


*Case I—Omega Model*


(iii-a) The regression Equations (24b) and (25b) were used to determine creep parameter *A*, and *n* was the stress exponent for the Omega model [50].


*Case II—Kachanov–Rabotnov model*


(iii-b) The Equations (24b) and (25b) were modified for the Kachanov–Rabotnov model in order to derive constants in the equation and by introducing the damage evolution parameter *ω* as follows in Equations (26) and (27):(26)$$n=\frac{n^{\prime }\sum \left(\ln(\frac{\sigma }{1-\omega })\ln\dot{\varepsilon }\right)-\sum \left(\ln\dot{\varepsilon }\right)\sum \left(\ln\frac{\sigma }{1-\omega }\right)}{n^{\prime }\sum [\left(\ln\frac{\sigma }{1-\omega }{)}^{2}\right]-(\sum \ln\frac{\sigma }{1-\omega }{)}^{2}}$$
(27)$$A=e^{\frac{\sum \left(\ln\dot{\varepsilon }\right)-n\sum \left(\ln\frac{\sigma }{1-\omega }\right)}{n^{\prime }}}$$

Figure 2 illustrates the strain rate against stress plots for the Omega and Norton–Bailey models obtained from the curve-fitting procedures. It was observed that curve fitting was achieved accurately for the given stress–strain values. The power law regression provides an accurate fitting, especially for the exponential data [39]. The coefficient of determination $R^{2}$ value, which quantifies the best-fitted plot and which is used effectively as an acceptance criterion for the prediction plot, was found to be 0.9803, which is well above the acceptance criterion for this plot as per ASME FFS-1/API-579 standards.

Again, Figure 3 depicts the strain rate against stress plots for the Omega-Kachanov–Rabotnov to Norton–Bailey models, which was obtained from the curve-fitting procedures by taking damage evolution parameter *ω* as 0.05. Damage evolution parameter *ω* or the tertiary creep damage constant was introduced to determine the effects of material damage at the tertiary stage for SS-316 material. The coefficient of determination $R^{2}$ value was found to be 0.99.

### 3.3. Development of Model in Finite Element Analysis

The dog bone specimen FEA model was developed based on ASTM ISO 204 & ISO R/206 E-139 standards for tensile creep testing in Abaqus [51]. The FEA model geometry was developed with the help of the sketcher toolbox in the Abaqus solver. Material properties for creep and plasticity were employed by section assignment. Similar, boundary conditions of the experiment were applied to the model. In the analysis step, maximum number of increments was set to 1000, with an increment size of 30,000 h. The reference point was selected for the direction of model displacement when the load applied. An appropriate mesh size was selected for proper convergence of the specimen. The Norton–Bailey model available in the Abaqus material library was employed for modelling the creep. Given the assumptions made for the FEM assessment, SS-316 isotropic material was selected for the analysis. A uniaxial load was applied on the specimen under defined boundary conditions and a thermal field. Material properties remained the same, irrespective of the load applied in any direction. An elastic perfectly plastic model was selected for plasticity and permanent plastic deformation. One end of the specimen was fixed, and the displacement was set at the other end with an amplitude of 2 mm/min. The thermal environment was created by applying predefined thermal fields throughout the model, with the temperature ranging from 0 °C–700 °C for the material SS-316 constant through the region [52]. Once the target temperature was obtained, a continuous longitudinal load was applied, resulting in the material’s grain structure dislocation and distortion. The load was maintained for the period of the test or until the specimen ruptured with a pre-stress of 35% UTS [53]. During the test, the data were continuously monitored and recorded to qualify for the stability of the temperatures, the load, and the specimen’s elongation [51]. The model was developed based on ASTM standards and as per the dimensions mentioned in Figure 4a. An example of the sample photo is presented in Figure 4b.

The physical properties for the isotropic material SS-316, Young’s modulus, and Poisson’s ratio were taken from ASME BPVC section II part D, sub-part 2 standards [44] for elasticity, which were temperature dependent. The values are depicted in Table 2. It is to be noted that there was a continuous decrease in Young’s modulus as the temperature of the material rose. The data for yield stress and plastic strain for plasticity were extracted again from ASME BPVC section II part D, sub-part 2 standards with respect to temperature [44]. As per the graph in Figure 5, it was observed that the material’s yield strength reduced as the temperature rose during the simulated creep test. The ASME standard values were devised after rigorous examination of the material behavior during exposure to varying temperatures and pressures and at different operating conditions.

Table 3 values, the creep parameter (*A*), and the stress exponent (*n*) were calculated by curve fitting for damage evolution through regression analysis for the Kachanov–Rabotnov model to the embedded Norton–Bailey creep law for SS-316 material. The values were obtained by keeping in view the effects of tertiary stage creep and the material behavior until rupture.

The damage paramter *ω* values were taken arbitrarily with the range from 0.05 to 0.40, as aligned with the work done by Christopher et al. [45], at a fixed temperature of 650 °C. The plot in Figure 6 displays the effects of damage evolution *ω* on the constants, the stress exponent, and the creep parameter of SS-316 material in the tertiary stage of creep at a fixed temperature of 650 °C. It is estimated that the rupture due to creep in the material usually occurs when the damage parameter reaches unity. From the graph it is clear that by increasing the value of damage parameter *ω*, there was a continuous decrease in the creep parameter (*A*), whereas the stress exponent (*n*) varied with respect to the rising damage parameter values.

After running the simulations, with pre-defined boundary conditions, the model was validated with the creep experimental test as per Christopher et al. [45]. The comparative assessment of the results was made among three creep models, Omega, Norton–Bailey and Kachanov–Rabotnov models, based on the proposed curve-fitting technique. The data were optimized later by using response surface methodology and the ANOVA technique to further analyze the results and for sensitivity studies of creep models for better damage prediction.

### 3.4. Sensitivity Analysis Using RSM and ANOVA

In this case study, a response surface methodology (*RSM*) model for creep strain was developed by using design expert software version 13 and by analysing simulation results obtained from Abaqus for the model. The corresponding design matrix, considering four independent design factors, which are represented as A: stress, B: stress exponent, C: creep parameter, and D: damage parameter, and one response: strain were analyzed. The response variable, strain, had a non-linear relationship with the independent variables, which happened to be a source of non-linearity in the selected model. In this case, a quadratic model was selected for best describing the relationship between the independent factors and the response variable. The design equations were solved to insert the data in the relevant slots of the design matrix for the response. Based on the statistical evaluation, the quadratic model was found to be significant, with the value of the coefficient of determination *R*^2^ approaching 0.80. After the model selection, 3D surface plots illustrating the correlation between design factors and responses were analyzed [54]. These plots were used to comprehend the behavior of all the responses. Eventually, the optimization criteria for each design parameter were specified with the desired degree of importance. The subsequent surface plots indicated the optimum values of design parameters.

The analysis of variance (ANOVA) statistical technique was also applied to determine the difference between two or more means or variables through significance tests, as it provides a way to make multiple comparisons of several population means [55]. After assigning low and high levels of designated factors, the design matrix comprised of 28 simulation runs was generated [56]. This matrix also included replicates of the central points in the attempt to have more reliability in the design and the analysis, as depicted in Table 4 [57].

### 3.5. Model Validation

The proposed model based on the curve-fitting technique was validated by the actual experimental creep test conducted by Christopher et al. [45]. The results showed good agreement between the predicted and the actual creep strain results. The predicted FEA creep strain was obtained between stress values ranging from 100 to 200 MPa at various time increments, with pre-defined boundary conditions and at a fixed temperature of 650 °C. The creep strain and creep strain rate were compared with the actual creep experiments [45]. There was a good agreement between the simulated and the experimental creep strain results at various stresses, as shown in the combined Figure 7. Because the deformation in the primary creep stage was negligible, it can be ignored.

Damage evolution parameter *ω* values were taken arbitrarily in the range from 0.05 to 0.40, as the tertiary stage creep damage was constant for SS-316 material, in order to track the behavior in that creep stage. The *ω* value helps in predicting the minimum creep strain, the creep deformation, and the rupture of the material. The graph in Figure 8a was plotted for different *ω* values for predicted creep strain versus time while running the simulation. It shows good agreement with the graph in Figure 8b [45] for various *ω* values, thus tracking material behavior in the tertiary stage of creep deformation.

The ratio between the strain rate and the damage rate was obtained and plotted against damage variables to show the impact of the damage rate on the strain rate. From the graphs, the dominance of the damage rate was visible, which is in good agreement for the simulation and the experimental results in the combined Figure 9 [45]. Thus, the model based on the proposed curve-fitting technique was validated for the analysis.

## 4. Results

### 4.1. Dog Bone Specimen Simulation

Von Mises stresses were developed in the specimen after running the simulation for 30,000 h under the defined boundary conditions for creep. Von Mises stresses were obtained by selecting the point at the center in the red zone of the specimen where high creep strain occurred. It was observed that the induced stresses were reduced gradually as the time progressed from 200 MPa to around 100 MPa, which was until the completion of the visco-elastic plastic run time of 30,000 h and as per the graph in Figure 10a. The relaxed stresses are the observed decrease in stresses, in response to the strain generated in the specimen. As per the graph in Figure 10b, the relaxed stresses were developed as the dog bone specimen endured creep strain due to permanent plastic deformation and rupturing

The deformation was apparent as the load was applied continuously, exceeding the material’s yield stress and the ultimate tensile strength leading to fracture. The model was under uniaxial tensile load to exhibit creep phenomena for SS-316 material. The effect of the deformation was obvious at the lower end of the specimen, which was fixed by applying symmetric boundary conditions. Figure 11a depicts the creation of a thermal environment with boundary conditions, fixing one end and with the load applied at the other end with meshed geometry. Figure 11b shows the Von Mises stress distribution on the specimen after running the simulation up to the defined time period of 30,000 h. The concentration of the stresses was at a maximum at the center and then concentrated around the fixed end of the specimen; the lower stresses were induced at the free end of the specimen. As the load was applied the deformation started as soon as the material entered from the elastic to the plastic region under the influence of the thermal environment and with the defined boundary conditions.

### 4.2. Creep and Plastic Strain Initiation and Propagation

The specimen underwent creep deformation, as depicted in Figure 11c, leading to material rupture. Due to the continuous loading, the material surpassed its yield strength and ultimate tensile strength limits, leading to material rupture. The deformation was initiated and then propagated throughout the model, as the constant uniaxial load was applied at one end of the specimen, resulting in model displacement from the other end. It was observed that the amount of plastic strain accumulation was dependent on the load applied, the boundary conditions, and the thermal environment, which was maintained throughout the analysis. In this analysis, the elastic perfectly plastic model was employed for the plasticity. Fracture strain occurred at the surface starting from the center of the specimen initiating with necking. The creep deformation occurred by atoms’ dislocation and due to grain boundary sliding. It was observed that the material behavior was governed by the type of the material, the thickness of the sample, the sample size, and the amount of the load applied.

The total inelastic strain was obtained by super-imposing the creep and the plastic strains. Figure 12 exhibits the comparison plot for total inelastic strain, creep strain, and plastic strain. The creep strain remained almost zero initially but increased slowly for some time and then increased rapidly, until it finally became steady; the plastic strain exhibited the same behavior. The total inelastic strain was within the standard strain range values given in the ASME FFS-1/API 579-1 standards for the specific material SS-316 and the temperature. From the plot, it was clear that slow creep strain occurred in the beginning, which became consistent at secondary-stage creep deformation for the rest of the visco-elastic plastic run time of 30,000 h until the material ruptured.

### 4.3. Omega, Norton–Bailey, and Kachanov–Rabotnov Models Comparison

Comparison of creep strain rates vs time at a temperature of 650 °C for SS-316 material between Norton–Bailey and Kachanov–Rabotnov–Norton–Bailey regression models exhibited good agreement with some discrepancies, as per the graph in Figure 13. However, results obtained from the Omega model showed more deviation than the other two models. It is evident that creep strain rates followed the same path for all three cases initially, but, beyond 5000 h, the results started to deviate slowly. By increasing the sample size of the creep data with more stress points, a better curve fit may be achieved for predicting the material’s creep behavior accurately. With the identification of the tertiary creep stage constant, the damage evolution parameter *ω* in the Kachanov–Rabotnov model, a complete creep curve can be obtained for this model representing the three creep stages of the material.

Similarly, the plot in Figure 14 for the creep strain versus time for the analytical Omega, Norton–Bailey and Kachanov–Rabotnov regression models, at an elevated temperature of 650 °C for the specimen SS-316, followed similar curve paths. After 5000 h, they had started to deviate into the material’s secondary creep deformation stage. It was observed that most of the time spent by the material was in the secondary stage of creep, under continuous loading with slow and steady deformation. It was also noticed that the decrease in stress levels resulted in a decrease in the creep strain and an increase in thecreep rupture life.

### 4.4. Data Optimization by Statistical Modelling

In this study, a quadratic regression model was selected for the response analysis: strain. The appropriateness of the chosen regression models was validated through the regular coefficient of determination (*R*^2^), the adjusted coefficient of determination (adjusted *R*^2^), and the predicted coefficient of determination (predicted *R*^2^). The fit statistics for the response strain are enlisted in Table 5 and were obtained from a central composite design. The significance of the model is evident from the values of *R*^2^ (0.76), the adjusted *R*^2^ (0.52), the predicted *R*^2^ (−0.94), and the adequate precision (9.78). Moreover, the values of adjusted *R*^2^ and the predicted *R*^2^ were in proximity to each other. Adequate precision compares the predicted values, called signal, and the average prediction error, called noise. The appropriate relationship between signal and noise is confirming the effectiveness of the model [58].

Table 6 depicts the model summary statistics for the response strain. The *F* and *p* values are indicating that the model is suitable for searching the design space, under the given conditions.

To verify the adequacy of the developed model, the predicted versus actual responses were plotted. Figure 15a shows the normal plot for residuals for the response strain. Figure 15b shows, graphically, the actual and predicted values for the response strain. The predicted and actual values for the concerned response, as depicted by the graph, were in close agreement with each other. The distribution of data points along the run order suggests that there was no significant increase or decrease in the values predicted by the model [59].

Most of the values were in close proximity to the central line with random scattering. No specific pattern of residuals above and below the central line was observable, in this case, which leads to the conclusion that the run order of the design process had little influence on the data; therefore, the model is significant [60].

ANOVA was used to determine the statistical parameters and the synergistic effects of each element. Various ANOVA adequacy tests (the lack-of-fit test, the *F*-value, and the *p*-value) recommend the regression model’s applicability and suitability. Table 7 shows the results of a statistical analysis using ANOVA to establish the degree of significance for thechosen model based on the agreement between anticipated and experimental/simulated responses.

The 3D model evaluated the response’s behaviour and demonstrated the independent factors’ synergistic impacts on the chosen response. Generally, the 3D model represents each response as a function of two independent factors where the remaining two factors are kept constant at their mean coded values. The interrelationship between the factors and the responses in RSM are best described by surface plots. A three-dimensional surface plot shows the functional relationship between the designated dependent variables and the corresponding independent variable. The response surface plot shows the combined effects of the variation of the stress exponent, the creep parameter, the stress and the damage parameter on the individual response strain, as in Figure 16a and the contour creep deformation map in Figure 16b.

### 4.5. Discrete Effects of Factors on the Response

The optimization ramps for design parameters are shown in Figure 17. The criteria defined for optimizing the responses are depicted by each ramp. The value of one of the independent variables was bound to vary within the specified range, while the other was set to be maximized. The red pointer indicates the optimum value for each factor, while the blue one disseminates the same information about the response [61].

The perturbation graph is an essential diagram depiction to analyse the influence of all variables at a given location in the design space. Figure 18 shows the plot of perturbation of the four variables on the strain response. It was observed that the parameter *A*: stress had a maximum effect on the response: strain, whereas the creep parameter: *C* had the least effect on the response. The other two parameters, the stress exponent: *B* and the damage parameter: *D*, had nominal effects on the response strain [62].

## 5. Conclusions

The significance of the study conducted was to curve-fit damage evolution parameters through regression analysis. The technique was suitable in analyzing the tertiary creep damage behavior of any material. The creep parameters obtained by curve fitting are vital, as they are required as inputs while defining material properties in the FEA package Abaqus. The problem with the creep tests include the fact that they are expensive and take a long time to complete; therefore, there is always a scarcity of creep data and results for any material. The following conclusions were made from this research study:(1)The method formulated in this article can be applied to curve-fit tertiary creep damage evolution parameters and to run creep analysis by finite element methods for any material. By deriving a damage evolution constant in the equation, a material’s behavior in the tertiary stage can be identified and predicted. A complete creep curve can be obtained covering all three stages by applying this technique.(2)From the results it is clear that the Omega model can work as a tool, because creep strain analytical data can be extracted from ASME FFS-1/API-579 standards and applied to embedded Norton–Bailey and Kachanov–Rabotnov models by regression analysis in Abaqus for any material. Obtained creep parameters work as inputs along with other parameters in the FEA package for damage evolution. Comparative assessment for creep strain was made among the Omega, Kachanov–Rabotnov, and Norton–Bailey models based on the proposed curve-fitting technique.(3)The fit statistics for the quadratic model of creep strain points revealed that the anticipated and simulated/actual values were more closely aligned. This proved that the quadratic model could navigate the design space effectively. Furthermore, as evidenced by their *p*-values, the interaction terms of mixing conditions had a substantial influence on the variables and the response.(4)Detailed statistical analysis and successive geometric optimization were performed using the response surface modelling approach and the ANOVA technique. The resulting 3D surface plot was analysed to comprehend the combined effect of the design factors: stress, the stress exponent, the creep parameter, and the damage evolution parameter on the relevant response: strain. The impact on the strain response was analysed and investigated with the help of contour creep deformation maps.(5)The FEA model was validated with the published experimental creep test, and the results showed good agreement between simulated and experimental results. Hence, the model was validated and applied. The combined effects in uncertainties can be removed by increasing the sample size of the creep data and for further extrapolation for creep prediction.

## Figures and Tables

**Figure 1 materials-14-05518-f001:**
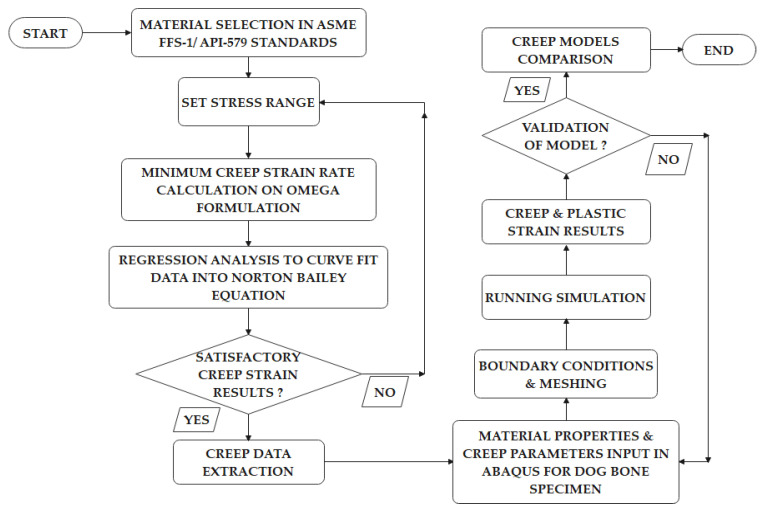
The overall process flow of the methodology to obtain creep data based on the Omega-Norton–Bailey regression model for the Abaqus dog bone specimen.

**Figure 2 materials-14-05518-f002:**
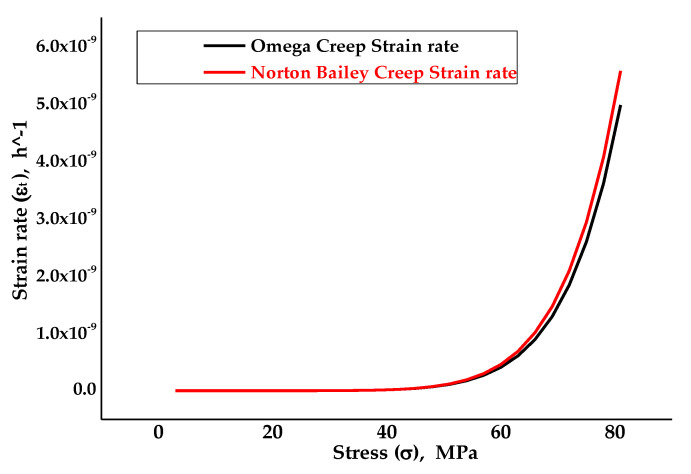
The resultant strain rate vs stress curve obtained following the curve fitting of MPC Omega to the Norton–Bailey model based on strain hardening [39].

**Figure 3 materials-14-05518-f003:**
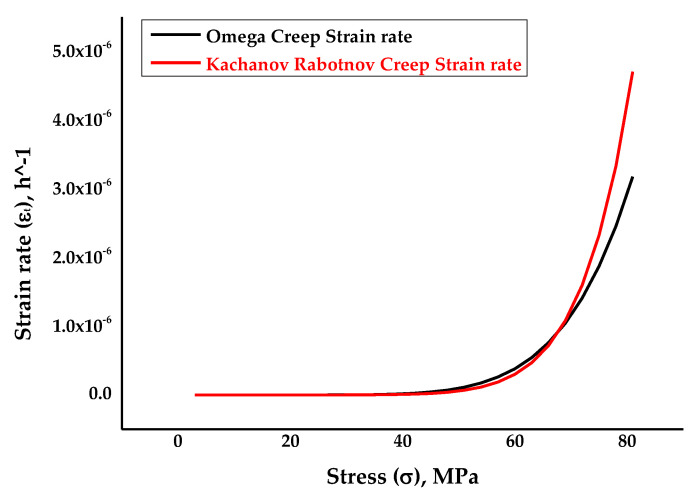
The resultant strain rate vs stress curve obtained following the curve fitting of MPC Omega to Kachanov–Rabotnov–Norton–Bailey model at *ω* = 0.05 [39].

**Figure 4 materials-14-05518-f004:**
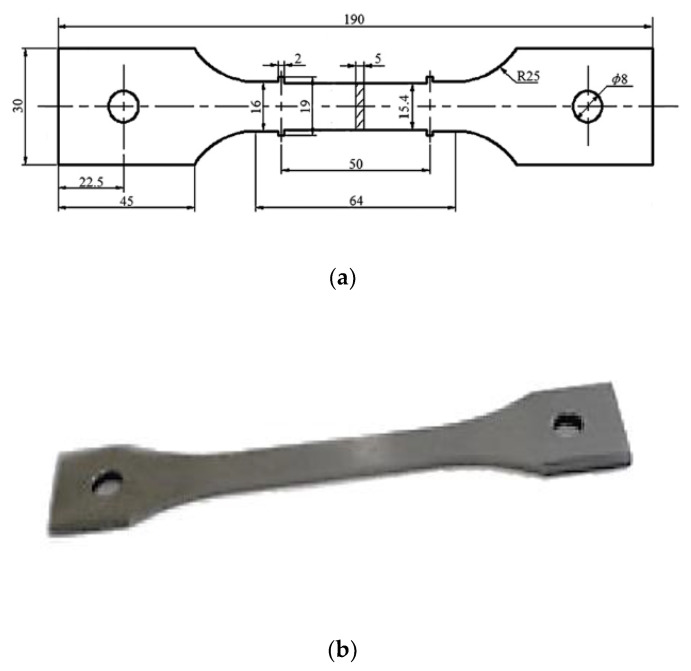
(**a**) Test specimen geometry for the creep test in FEA. (**b**) Sample photo of specimen [51].

**Figure 5 materials-14-05518-f005:**
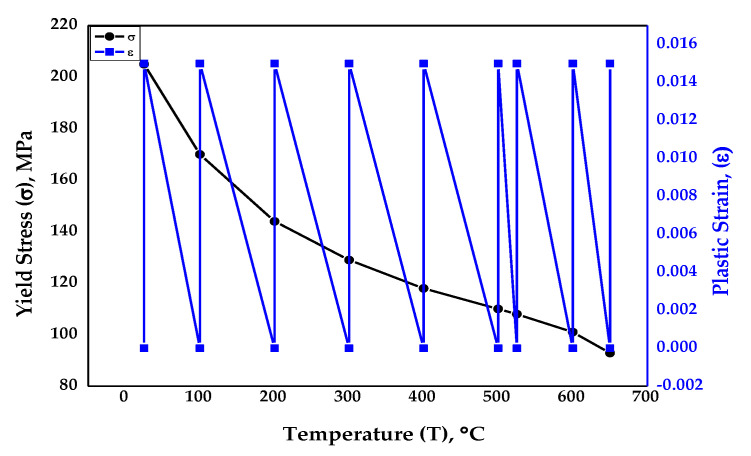
Material properties of isotropic material SS-316 [44].

**Figure 6 materials-14-05518-f006:**
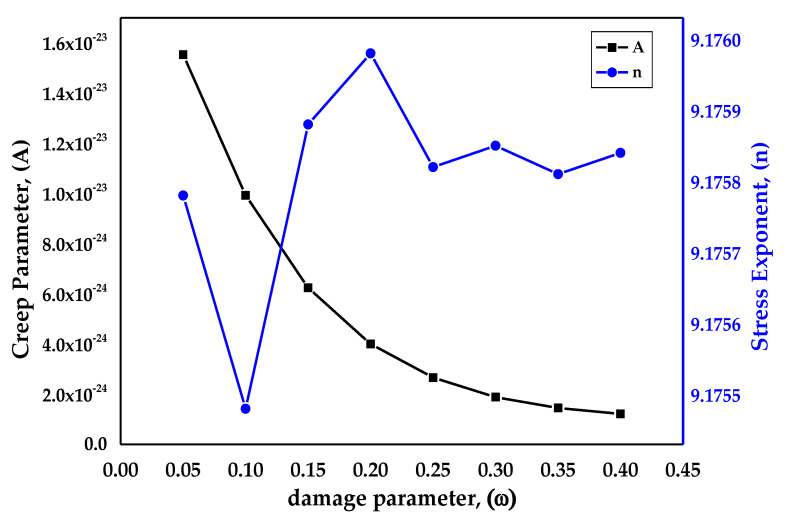
Effects of damage evolution on creep parameters at 650 °C.

**Figure 7 materials-14-05518-f007:**
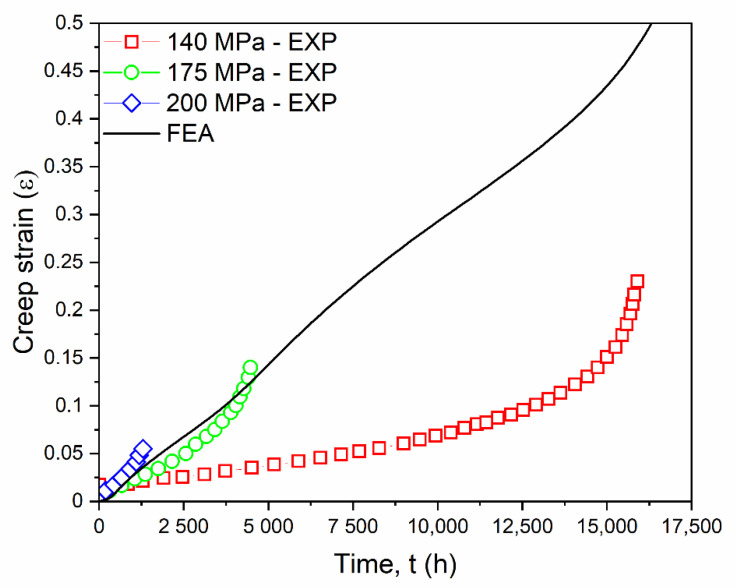
Comparison of predicted creep strain by FEA simulation with experimental creep strain at 650 °C [45].

**Figure 8 materials-14-05518-f008:**
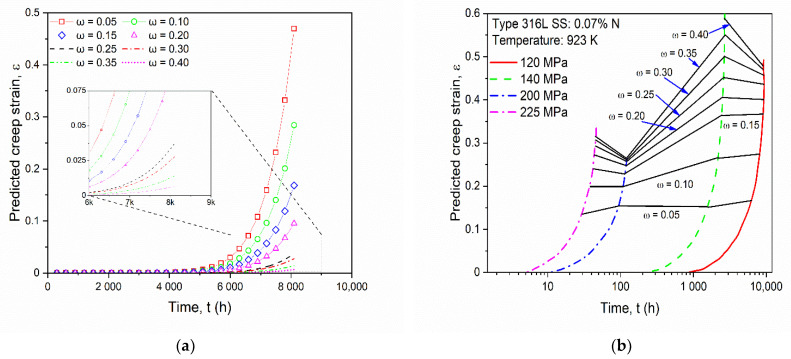
(**a**) Predicted creep strain at different *ω* values for SS-316 material. (**b**) Creep strain results at various damage evolution parameters from the reference [45].

**Figure 9 materials-14-05518-f009:**
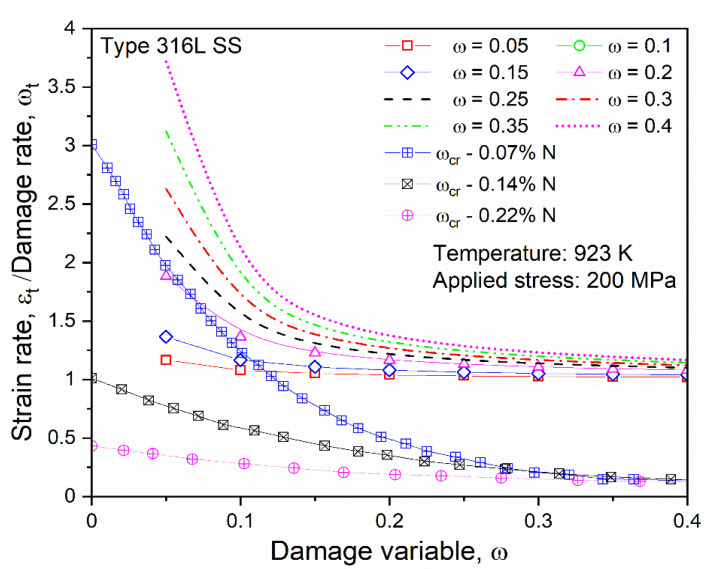
Comparison of the variations in the ratio between the strain rate and the damage rate for the simulation and the experiment [45].

**Figure 10 materials-14-05518-f010:**
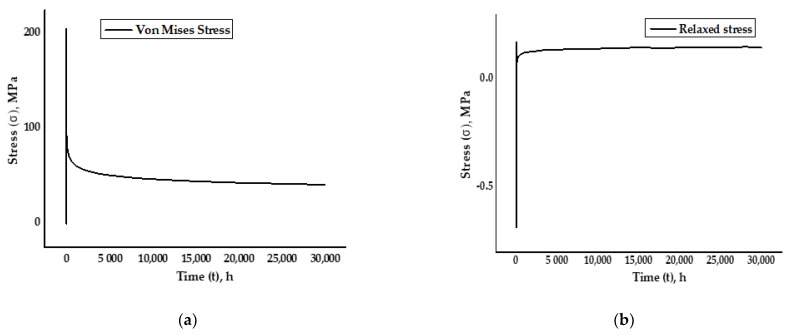
(**a**) Von Mises stress distribution. (**b**) Relaxed stress with the visco-elastic plastic run time of 30,000 h.

**Figure 11 materials-14-05518-f011:**
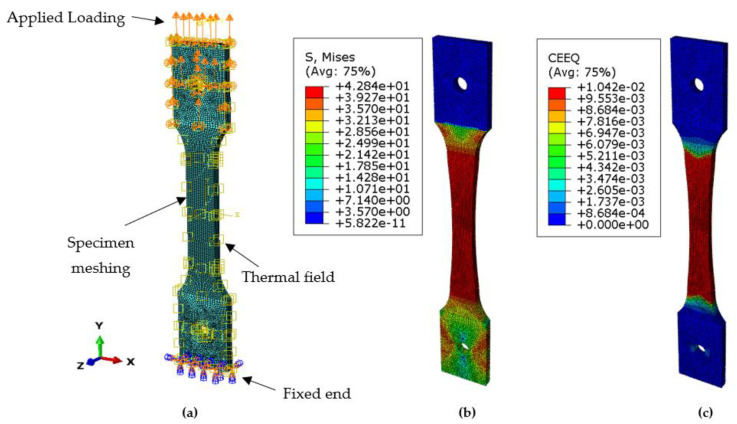
(**a**) FE model of specimen. (**b**) Induced Von Mises stress in the specimen after running simulation. (**c**) Creep strain (CEEQ) for the applied stresses.

**Figure 12 materials-14-05518-f012:**
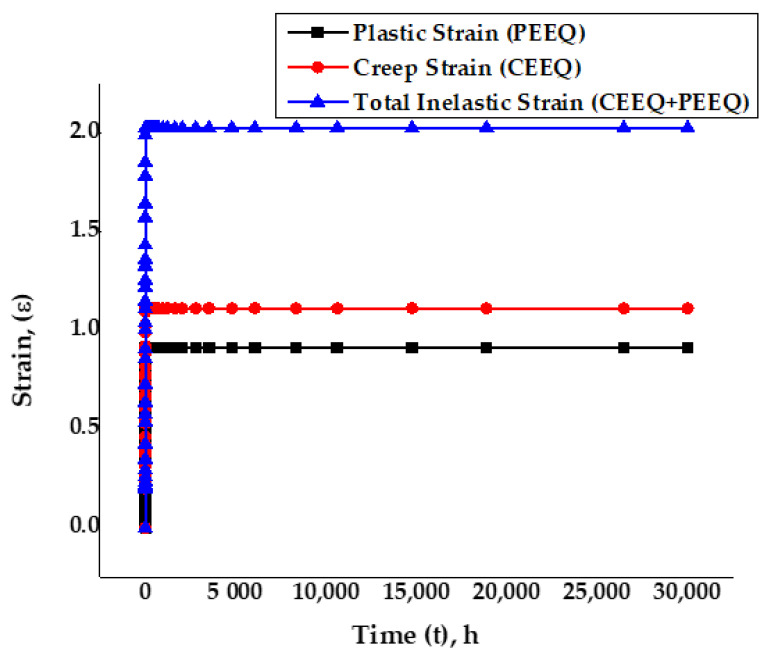
Accumulated creep strain and plastic strain for the dog bone coupon SS-316.

**Figure 13 materials-14-05518-f013:**
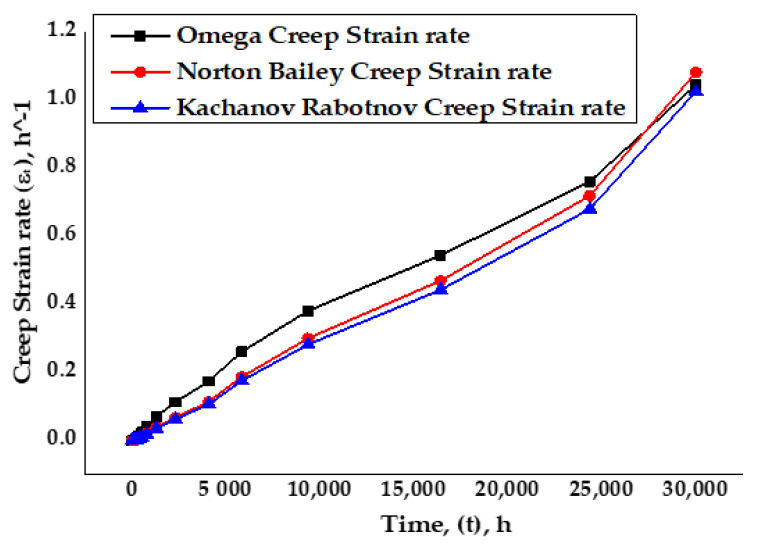
Analytical Omega model vs Norton–Bailey and Kachanov–Rabotnov creep strain rates.

**Figure 14 materials-14-05518-f014:**
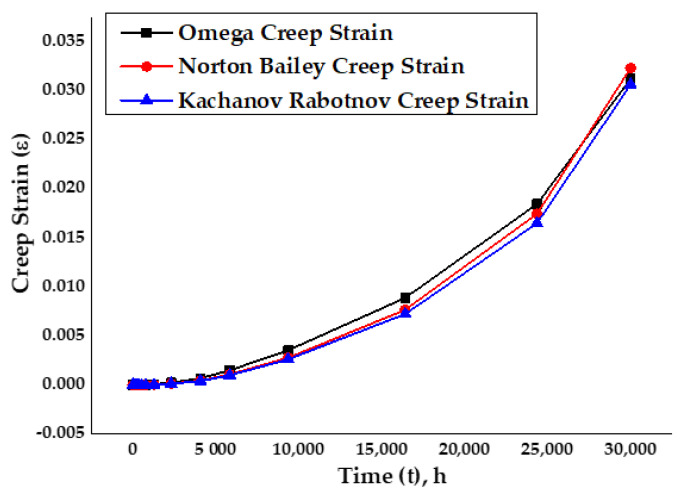
Omega model analytical vs Norton–Bailey and Kachanov–Rabotnov creep strains.

**Figure 15 materials-14-05518-f015:**
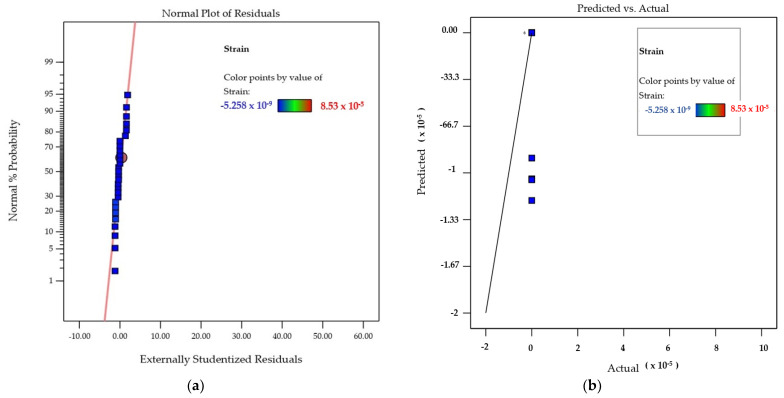
(**a**) Normal plot of residuals for the response strain. (**b**) Actual and predicted values for the response strain (*ε*).

**Figure 16 materials-14-05518-f016:**
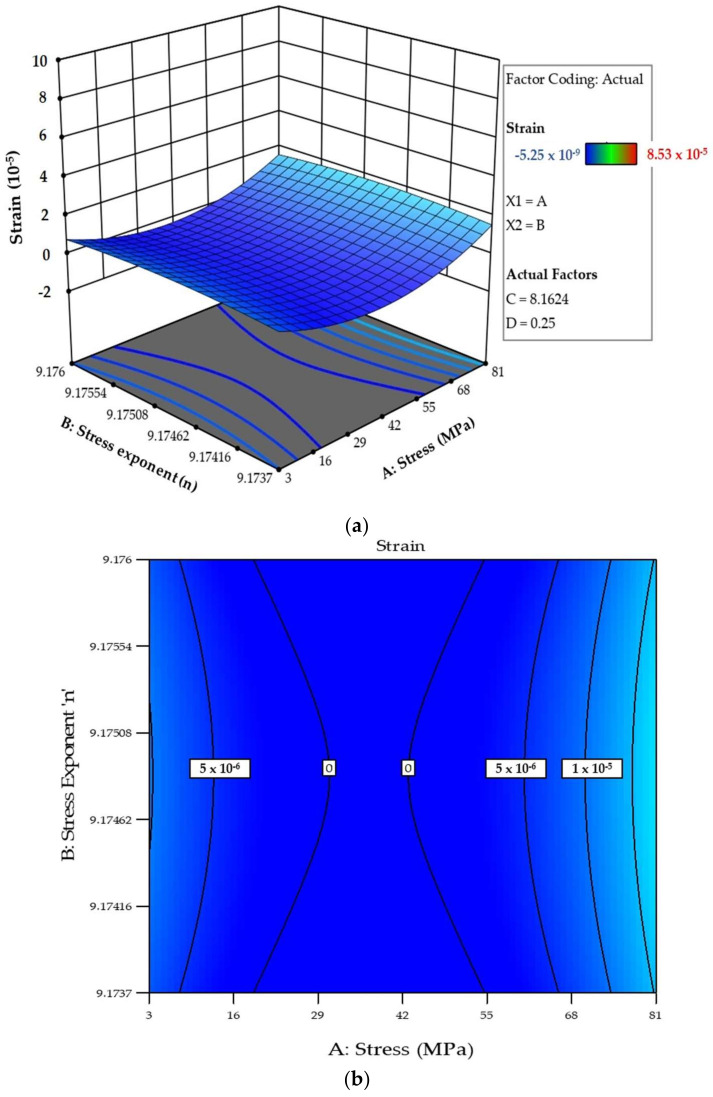
(**a**) The combined effect of design factors on the response strain (*ε*), (**b**) contour creep deformation map.

**Figure 17 materials-14-05518-f017:**
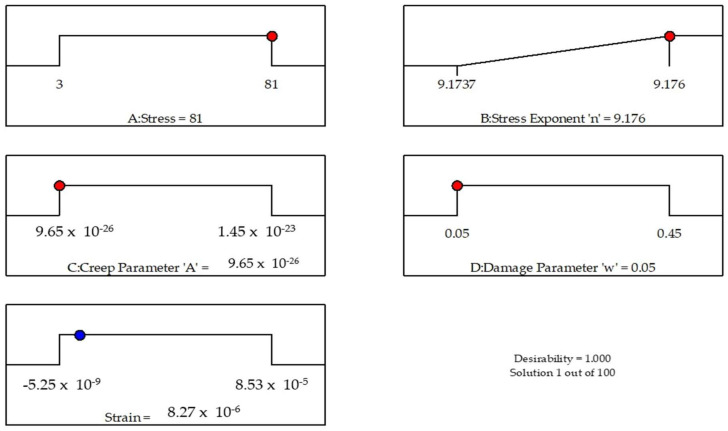
Optimization criteria for factors and response.

**Figure 18 materials-14-05518-f018:**
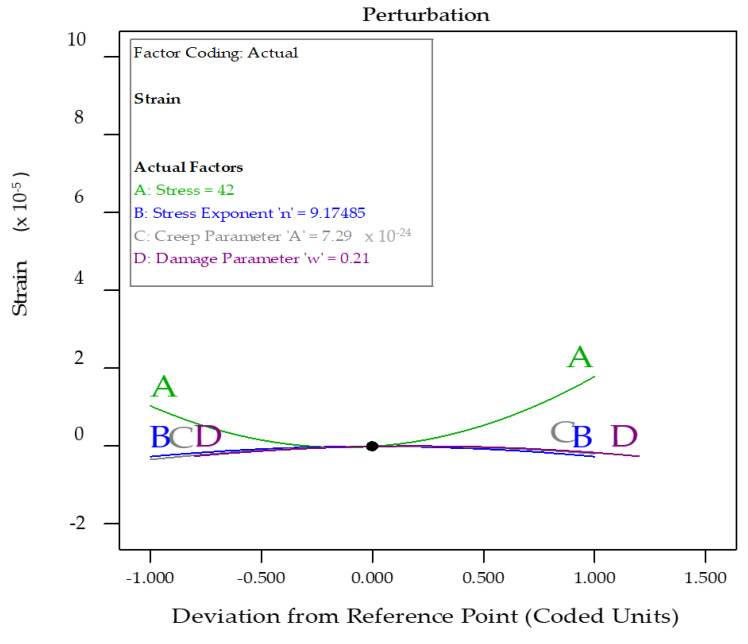
Pertubation plot depicting the effects of all variables simultaneously with a central reference point on strain.

**Table 1 materials-14-05518-t001:** MPC Omega model material coefficients for material type SS-316 (MPa, °C) [43].

$\mathrm{Strain}\;\mathrm{Rate}\;\mathrm{Parameter}—\varepsilon_{c0}$		Omega Parameter—*Ω*	
*A* _0_	−18.9	*B* _0_	−4.163
*A* _1_	41,230.11	*B* _1_	16,793.192
*A* _2_	−12,446.783	*B* _2_	−10,221.744
*A* _3_	1299.221	*B* _3_	1634.960
*A* _4_	111.222	*B* _4_	222.222

**Table 2 materials-14-05518-t002:** Physical properties for SS-316 at selected temperatures [44].

Young’s Modulus (MPa)	Poisson’s Ratio	Temperature (°C)
134,000	0.31	−25
128,000	0.31	65
120,000	0.31	100
115,000	0.31	125
111,000	0.31	150
104,000	0.31	200
97,600	0.31	250
93,100	0.31	300
90,700	0.31	325
88,400	0.31	350
86,600	0.31	375
84,700	0.31	400
83,500	0.31	425
82,300	0.31	450
80,500	0.31	475
79,100	0.31	500
77,800	0.31	525
76,800	0.31	550
74,700	0.31	575
70,000	0.31	600
55,300	0.31	625
42,900	0.31	650

**Table 3 materials-14-05518-t003:** Material SS-316 constants for creep at temperatures 630–675 °C, *ω* = 0.05.

Creep Parameter *A*	Stress Exponent *n*	Temperature °C
1.71460 × $10^{-24}$	9.37430	630
2.96370 × $10^{-24}$	9.32270	635
5.09125 × $10^{-24}$	9.27170	640
8.70190 × $10^{-24}$	9.22120	645
1.47830 × $10^{-23}$	9.17100	650
2.48800 × $10^{-23}$	9.12180	655
4.19040 × $10^{-23}$	9.07300	660
7.00796 × $10^{-23}$	9.02400	665
1.1612 × $10^{-22}$	8.97670	670
1.9187 × $10^{-22}$	8.92940	675

**Table 4 materials-14-05518-t004:** Design matrix of factors and response.

Std	Run	Factor 1 A: Stress, *σ* (MPa)	Factor 2 B: Stress Exponent, *n*	Factor 3 C: Creep Parameter, *A* (MPa^−n^ h^−1^)	Factor 4 D: Damage Parameter (*ω*)	Response: Strain (*ε*)
29	1	42	9.17485	7.29825 × 10^−24^	0.25	5.40131 × 10^−9^
3	2	3	9.176	9.65 × 10^−26^	0.05	1.97521 × 10^−21^
26	3	42	9.17485	7.29825 × 10^−24^	0.25	5.40131 × 10^−9^
21	4	42	9.17485	2.17017 × 10^−23^	0.25	1.6061 × 10^−8^
5	5	3	9.1737	1.45 × 10^−23^	0.05	2.96056 × 10^−19^
23	6	42	9.17485	7.29825 × 10^−24^	0.65	4.94469 × 10^−9^
15	7	3	9.176	1.45 × 10^−23^	0.45	7.79436 × 10^−20^
25	8	42	9.17485	7.29825 × 10^−24^	0.25	5.40131 × 10^−9^
17	9	120	9.17485	7.29825 × 10^−24^	0.25	8.53267 × 10^−5^
14	10	81	9.1737	1.45 × 10^−23^	0.45	4.43651 × 10^−6^
12	11	81	9.176	9.65 × 10^−26^	0.45	2.98253 × 10^−8^
24	12	42	9.17485	7.29825 × 10^−24^	0.25	5.40131 × 10^−9^
9	13	3	9.1737	9.65 × 10^−26^	0.45	5.17613 × 10^−22^
10	14	81	9.1737	9.65 × 10^−26^	0.45	2.95257 × 10^−8^
13	15	3	9.1737	1.45 × 10^−23^	0.45	7.7776 × 10^−20^
18	16	42	9.17255	7.29825 × 10^−24^	0.25	5.35515 × 10^−9^
19	17	42	9.17715	7.29825 × 10^−24^	0.25	5.44787 × 10^−9^
28	18	42	9.17485	7.29825 × 10^−24^	0.25	5.44787 × 10^−9^
27	19	42	9.17485	7.29825 × 10^−24^	0.25	5.44787 × 10^−9^
4	20	81	9.176	9.65 × 10^−26^	0.05	3.12123 × 10^−8^
1	21	3	9.1737	9.65 × 10^−26^	0.05	1.97031 × 10^−21^
6	22	81	9.1737	1.45 × 10^−23^	0.05	4.64277 × 10^−6^
11	23	3	9.176	9.65 × 10^−26^	0.45	5.18728 × 10^−22^
22	24	42	9.17485	7.29825 × 10^−24^	−0.15	5.89512 × 10^−9^
2	25	81	9.1737	9.65 × 10^−26^	0.05	3.08984 × 10^−8^
16	26	81	9.176	1.45 × 10^−23^	0.45	4.48152 × 10^−6^
7	27	3	9.176	1.45 × 10^−23^	0.05	2.96794 × 10^−19^
8	28	81	9.176	1.45 × 10^−23^	0.05	4.68992 × 10^−6^

**Table 5 materials-14-05518-t005:** Fit statistics for strain (*ε*).

Statistical Parameters	Values	Remarks
*R* ^2^	0.7643	The quadratic model is significant to search the design space
Adjusted *R*^2^	0.5286	
Predicted *R*^2^	−0.9414	
Adequate Precision	9.7876	

**Table 6 materials-14-05518-t006:** Model summary statistics for response strain (*ε*).

Source	Sum of Squares	df	Mean Square	*F*-Value	*p*-Value	
Mean vs. Total	3.713 × 10^−10^	1	3.713 × 10^−10^			Suggested Aliased
Linear vs. Mean	1.679 × 10^−9^	4	4.198 × 10^−10^	1.90	0.1439	
2FI vs. Linear	2.056 × 10^−11^	6	3.427 × 10^−12^	0.0117	1.0000	
Quadratic vs. 2*FI*	3.654 × 10^−9^	4	9.112 × 10^−10^	7.74	0.0017	
Cubic vs. Quadratic	1.648 × 10^−9^	8	2.060 × 10^−10^	4.352 × 10^−9^	<0.0001	
Residual	2.84 × 10^−19^	6	4.734 × 10^−20^			
Total	7.364 × 10^−9^	29	2.539 × 10^−10^			

**Table 7 materials-14-05518-t007:** *ANOVA* for the quadratic model (response: strain, *ε*).

Source	Sum of Squares	df	Mean Square	*F*-Value	*p*-Value
Model	5.345 × 10^−9^	14	3.818 × 10^−10^	3.24	0.0176
*A*-Stress	2.428 × 10^−10^	1	2.428 × 10^−10^	2.06	0.1729
*B*-Stress Exponent *n*	3.601 × 10^−16^	1	3.601 × 10^−16^	3.059 × 10^−6^	0.9986
*C*-Creep Parameter *A*	1.376 × 10^−11^	1	1.376 × 10^−11^	0.1169	0.7375
*D*-Damage Parameter *ω*	7.326 × 10^−15^	1	7.326 × 10^−15^	0.0001	0.9938
*AB*	5.380 × 10^−16^	1	5.380 × 10^−16^	4.570 × 10^−6^	0.9983
*AC*	2.054 × 10^−11^	1	2.054 × 10^−11^	0.1745	0.6825
*AD*	1.089 × 10^−14^	1	1.089 × 10^−14^	0.0001	0.9925
*BC*	5.239 × 10^−16^	1	5.239 × 10^−16^	4.450 × 10^−6^	0.9983
*BD*	2.917 × 10^−19^	1	2.917 × 10^−19^	2.478 × 10^−9^	1.0000
*CD*	1.060 × 10^−14^	1	1.060 × 10^−14^	0.0001	0.9926
*A* ^2^	3.235 × 10^−9^	1	3.235 × 10^−9^	27.48	0.0001
*B* ^2^	1.824 × 10^−10^	1	1.824 × 10^−10^	1.55	0.2337
*C* ^2^	1.824 × 10^−10^	1	1.824 × 10^−10^	1.55	0.2337
*D* ^2^	1.824 × 10^−10^	1	1.824 × 10^−10^	1.55	0.2337
Lack of fit (LOF)	1.648 × 10^−9^	9	1.831 × 10^−10^	Insignificant LOF shows a good fit for the model	

## Data Availability

The data presented in this study are available on request from the corresponding author.

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
