# Peer review of "Curve Fitting for Damage Evolution through Regression Analysis for the Kachanov–Rabotnov Model to the Norton–Bailey Creep Law of SS-316 Material"

_materials, 2021, doi:10.3390/ma14195518_

Round 1

Reviewer 1 Report

The present article investigates the Damage Evolution through Regression Analysis for Kachanov-Rabotnov model to Norton Bailey Creep Law of SS-316 material. This is an interesting topic however the presentation of the manuscript can be better after some corrections as follow:

  1. The following articles discussed different existing models to curve fitting creep life of steel on elevated temperature. However a model that is proposed for steel sheets under creep condition is modified Theta projection model that authors missed to taken into consideration. I suggest you use the following references in your introduction.
  • Alipour, R., & Nejad, A. F. (2016). Creep behaviour characterisation of a ferritic steel alloy based on the modified theta-projection data at an elevated temperature. International Journal of Materials Research, 107(5), 406-412.

  • Alipour, R., Farokhi Nejad, A., & Nilsaz Dezfouli, H. (2018). Steady state creep characteristics of a ferritic steel at elevated temperature: an experimental and numerical study. ADMT Journal, 11(4), 115-129.

  1. Figure 5 needs to replace with a high-quality figure or merge with figure 4.
  2. The figures have to follow the same style. For example figures: 8, 9(a), 10(b) and 14 has to be changed with a single style.
  3. FE model development is not satisfactory and it should be explained more. The existing model in ABAQUS are modeling steady state creep region. The authors should explain how they obtained the results even in the rupture region. If they write a user-defined subroutine they need to clarify it.
  4. Figures 12 and 13 should be merge and redundant contour plots such as PEEQ and meshed pictures can be omitted. The meshed picture can be merged with applied boundary condition and loading figure and use labels to presenting better.
  5. Why author uses RSM method and what is the reason for avoiding factorial methods. Have the authors seen a nonlinearity source in the statistical model? It should be explained briefly
  6. In figure 17 only a single point is out of the predicted model and CF lines. It is maybe because of numerical error or other sources of testing error. You can remove this point from your ANOVA analysis and show normal distribution and other indicators with higher resolution.
  7. The discussion about statistical analysis is too much and some part is not necessary to be as the part of a scientific paper. The authors should rewrite this part of the manuscript.

Regarding the above-mentioned comments, the current manuscript is not acceptable for publishing and I look forward to receiving any revised version, with all the changes and additions made clearly highlighted.  

Author Response

'Please see the attachment'

Reviewer 2 Report

Very interesting topic. Please describe in more detail the assumptions for FEM analysis.
In line 204 an unnecessarily repeated "min"

Author Response

'Please see the attachment'

Reviewer 3 Report

Thank you for the opportunity to review the work. it is interesting and shows important issues. The theoretical introduction is extensive and covers the topic well and presents the current state of knowledge in the area of the article.
The very structure of the publication is also correct, after the introduction theoretical information and models appear, then analyzes to summarize the whole with correct conclusions based on the obtained research results.
However, it is worth introducing some modifications to the work in question:
- standardize the system of notation of units on the graph axes (e.g. Fig. 11 and Fig. 6 - once sigma and once MPa is in parentheses)
- standardize the writing of formulas (2, 6, 7, 8, 10, 11 - they are bold, the reserve is not)
- unnecessary characters in table 2

Author Response

'Please see the attachment'

Reviewer 4 Report

Sorry, I have to reject your paper. Please, engage a good friend for re-reading and improving your English. Your equations (24a) and (24b) are false.

Author Response

'Please see the attachment'

Round 2

Reviewer 1 Report

The present manuscript is satisfactory now and all corrections have been implemented. It is acceptable and can be published in the present form.

Author Response

Thanks for accepting the manuscript after corrections for publication.  

Reviewer 4 Report

Dear Authors,

I like it that you are fighting for your paper and hope that all will come to a good end.

At first a remark. I write often reports for scientific journals. I always trust the editor to have a sound paper which merits reviewing. In your case the main idea is interesting, probably even very good.

Under this basic assumption, I try first to get some impression
of the care used in writing the paper. My starting point is the list of references. Is it made with care? If not, I am negatively biased.

Let's look at your references.

[1] Who published this paper? Any source?
[3] You have here capital letters but not in [2]
[4] Seems to be incomplete.
[5] I miss an "and" between the names. And perhaps the reference is incomplete.
[7] This is right, because it is a book.
[8] Perhaps the names Kachanov and Rabotnov with capital letters
[12] This is a book, publisher probably McGraw Hill. What means the "no."?
[14] See [3]
Here I stopped and came to a negative bias.
By the way, which is the difference between your paper and [48]?

The next step is to consider the Abstract. How is its English, can I
understand it?

In your case it is fine until line 24. Then it is not clear for me: do you
speak about your paper or again about the literature. Perhaps write
instead of " The extrapolation..." "It extrapolates the creep behavior
by fitting the Kachanov-Rabotnov model to ..." To which data the model
is fitted? Your sentence is incomplete.
In line 32 I recommend to add after "creep test" the word "data".
Perhaps start line 34 with "The results show".

Then I browse through the introduction, looking for strange points.
In your case I found:

line 48: "that" should be "which" (and if "that", then no comma)
line 59: you explain again the abbreviation CTM
line 65: This sentence is totally isolated in your text.
line 79: A strange sentence. And until the end of the paragraph
         the text is also isolated from the rest.

Then I study the beginning of the exposition, of the theoretical
framework, assuming that the main idea of the paper is sound.

In eq. (2) and (5) I see some "'" above the "=".
line 212: Which equation is simplified to obtain (3)?
          Now you have epsilon_cr, before you had epsilon_min.
line 216: It seems that t = 1.
(5): With your A you have two sigmas in the equation.
line 220: unitless, not unit less.
line 221: I do not understand this sentence.
line 228: Shift the word "where" to the left-hand side.
line 230: Which equation collapses? I cannot believe that it is your
          (7) on the next page.

At this point I am close to giving up.

Two questions:
1. For me it is a rule to write all symbols in the text in italics,
as you have done it in the equations and in line 230. Why you do not follow always this rule?
2. Do you know that functions such as "ln" are always written not in 
italics, also in formulas?

And finally about the equations (24a) and (25a).Yes, these equations are closely related the classical linear regression formulas. However, in (24a) there you have two times "ln" in italics and in the denominator the 2 is not a factor but an exponent. And (25a) would be correct if i = n' and n = B.

Author Response

'Please see the attachment'

Round 3

Reviewer 4 Report

Dear authors,

your have improed your paper. Thank you. I recommend to

make the following changes in your Abstarct:

In a number...  computed --> studied,  discovered --> determined

trial-and-error

Author Response

'Please see the attachment'
